# Selection of Magneto-Optical Material for a Faraday Isolator Operating in High-Power Laser Radiation

**Ilya Snetkov [1,*] and Jiang Li [2,3]**

1 Federal Research Center Institute of Applied Physics, Russian Academy of Sciences, 46 Ulyanov Street, 603950 Nizhny Novgorod, Russia
2 Transparent Ceramics Research Center, Shanghai Institute of Ceramics, Chinese Academy of Sciences, Shanghai 201899, China
3 Center of Materials Science and Optoelectronics Engineering, University of Chinese Academy of Sciences, Beijing 100049, China
* Correspondence: snetkov@appl.sci-nnov.ru; Tel.: +7-831-416-06-74

**Abstract:** Faraday isolators are the inherent components of complex laser systems. The isolation degree is essentially determined by the effects that occur in its magneto-optical element, so the choice of material from which it is made is very important. The principal approaches to choosing a magneto-optical material for Faraday isolators are addressed. Characteristic features of materials for Faraday devices operating in laser radiation with high average and high peak power are considered. Some trends in magneto-optical ceramics and the advantages and shortcomings of a number of ceramic samples are analyzed. Using the proposed approaches and recommendations will allow to create devices with unique characteristics for any wavelength range for different practical applications.

**Keywords:** magneto-optical ceramics; Faraday isolators; thermal effects

## 1. Introduction

A dielectric permeability tensor (index of refraction and absorption coefficient) changes under the action of a magnetic field in most materials, which leads to different magneto-optical effects. One of such effects is a longitudinal Faraday effect-rotation of the polarization plane of linearly polarized radiation propagating along the lines of the magnetic field through an optical material placed in this field. This effect depends on the magnitude of the magnetic field, length of light propagation in the material, and on the characteristics of the material (the Verdet constant value dependent on the radiation wavelength and temperature). The Faraday effect is a nonreciprocal polarization effect that allows the controlling of radiation polarization, the intensity of polarized light together with the polarizer, tracing changes in the optical element length, changes in the magnetic field value, and changing the spectral composition of radiation. Therefore, this effect has found wide application for various manipulations with radiation, such as choppers, polarization switches, deflectors, displays [1], soft diaphragms [2], Q-switch modulators [3], narrow-band spectral filters [4], the remote sensing of current, magnetic field or even temperature in hostile environments [5], optical data storage [6], and others.

By virtue of the Faraday effect nonreciprocity, Faraday isolators (FI) and Faraday rotators (FR) have found wide application in laser physics. A traditional Faraday isolator is a magnetic system with a magneto-optical element (MOE) placed inside. The magnetic field and the MOE length are selected so that the rotation of the polarization plane is exactly 45°. The magnetic system is placed between two polarizers, the angle between the transmission planes of which is 45°. These devices are used to organize multi-pass amplifier schemes, schemes with compensation of thermally induced birefringence, as well as to provide optical isolation of one part of the scheme from another and to ensure the

propagation of laser radiation in a strictly specified direction. In this paper, the emphasis will be shifted towards the discussion of the choice of magneto-optical (MO) materials for free space bulk FIs for applications in laser radiation with high average and high peak power. Quite a lot of articles are devoted to the review of MO materials [7–10]. In contrast, in this work, less attention will be paid to specific materials, and more attention will be paid to the general principles of selection, and to the pros and cons of each of them.

To ensure a high isolation ratio when using traditional polarizers, the rotation of the plane of polarization in the MO material of the FI must be the same at all points of the cross section. Any deviations and inhomogeneities in the rotation of polarization will lead to a decrease in the isolation ratio of the device. Deviations may be caused either by transverse inhomogeneity of the field of the magnetic system, or with the material or processes that occur in the MOE during operation in laser radiation. Since the work is devoted to the choice of the MO material, we will exclude from consideration problems related to the magnetic field and assume that it is permanent and homogeneous in the MOE region. At low powers, polarization distortions are determined by the optical quality of the used MO material: inhomogeneities, inclusions, micro/macro strain and stresses, etc. These polarization distortions are linear in power and lead to a constant level of depolarization of the transmitted radiation referred to as "cold depolarization". With an increase in the radiation power, there arise thermally induced distortions of the refractive index, which significantly limit the performance of Faraday isolators.

Laser radiation parameters play a determining role in the choice of MO material for Faraday isolators. The material must be transparent for the wavelength of the used radiation to have a small absorption coefficient. The transmission spectra of a number of the studied MO materials are presented in Figure 1.

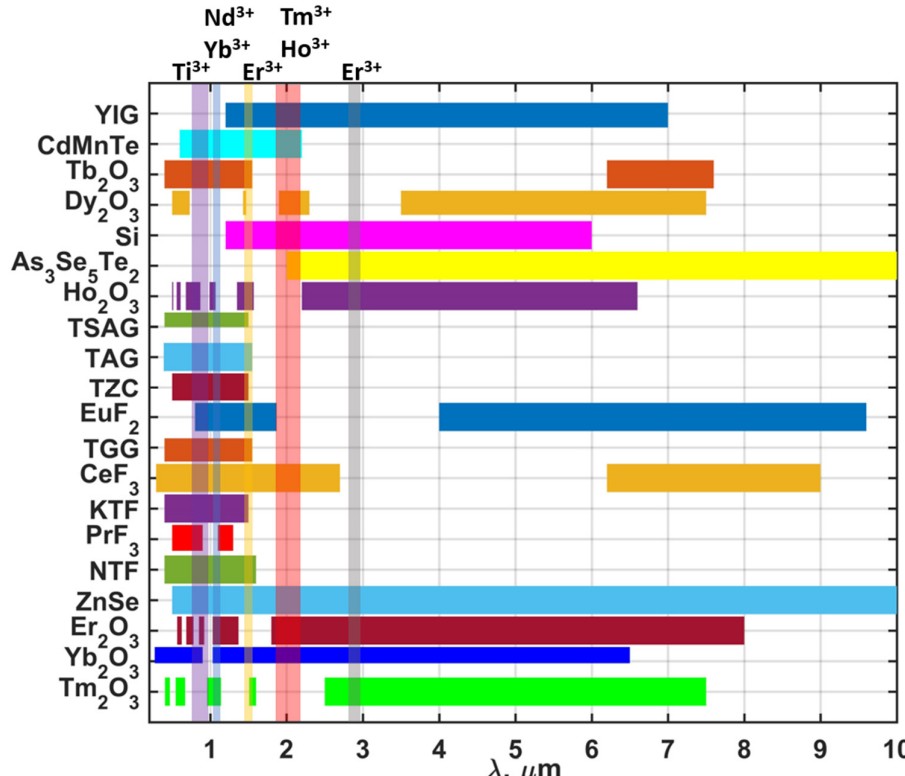

**Figure 1.** Transmission spectra of magneto-optical materials. Arranged in descending order of the Verdet constant value from top to bottom. Translucent vertical stripes indicate the characteristic spectral regions in which lasers based on the specified above ion can generate radiation.

The wavelength range of 0.7–1.1 μm is among the most widely used ones, which is associated with the presence of active ions with good laser properties and emitting in this range ($Ti^{3+}$, $Nd^{3+}$, and $Yb^{3+}$). The vast majority of technological and research lasers operate using active materials doped with these ions. In this wavelength range, MO materials containing terbium ($Tb^{3+}$) are among the best ones, as terbium has no absorption lines in the 0.5–1.5 μm range and possesses high magneto-optical properties. One of the most widely used materials is terbium gallium garnet (TGG), which has good magneto-optical ($V = 37$ rad/(T·m) $@\lambda = 1064$ nm), thermo-optical, and mechanical properties. Due to its wide prevalence and availability, it is convenient to use TGG as a reference material. Recently, there has been an increased interest in a longer wavelength range $\lambda = 1.5$–3 μm, in which active media doped with $Er^{3+}$, $Tm^{3+}$, and $Ho^{3+}$ ions emit (for example, the next generation of laser interferometers for detecting gravitational waves [11] or Big Aperture Thulium (BAT) laser [12]). However, terbium containing MO materials are not transparent for $\lambda = 1.5$–6 μm, and relatively few transparent materials with good magneto-optical and thermo-optical properties are known and their choice is significantly limited, which greatly complicates the development of FIs meeting ever-increasing requirements. In this work, we will consider promising MO materials for the $\lambda = 1.5$–3 μm range.

The peak and average radiation powers determine which of the effects have a stronger impact on deterioration of the FI isolation ratio: at a high peak power, nonlinear effects will limit the isolation [13], and at a high average power, thermal effects have the limiting effect [14]. For each FI, depending on the field of application, the corresponding requirements are set for the isolation ratio (in the operating mode of laser radiation), magnitude of possible power losses during passage through the FI, permissible deterioration of radiation quality (amplitude and phase modulation of the transmitted radiation), and permissible change in the radiation divergence caused by the thermal lens. Depending on the requirements and parameters of laser radiation, FI manufacturers choose a MO material and optical and mechanical schemes of the FI.

## 2. Magneto-Optical Materials of Different Types: Advantages and Shortcomings

Materials for MOE can be divided into several groups that differ in internal structure and have their own advantages and disadvantages: glass, single crystals, and ceramics. The technology of melting glass MOEs makes it possible to produce large-aperture elements with high optical quality (uniformity of the refractive index and Verdet constant, low scattering coefficient) relatively quickly and inexpensively. The technology is readily scalable and allows a relatively easy transfer to mass production. The vitrification process makes it possible to change the chemical composition of glasses within a fairly wide range and, as a consequence, to control their magneto-optical, thermo-optical, and nonlinear properties. Glasses can be made from various materials and, hence, transparent glass can be produced for the desired wavelength range. The optical properties of glasses are isotropic due to the amorphous structure, which simplifies any adjustments using glass MOEs, as well as an analitical description of most nonlinear and thermo-optical processes. The amorphous structure is also responsible for the main disadvantages of glass MOEs: low thermal conductivity rarely more than 1.3 W/(m K) (usually <1 W/(m K)); high fragility and lower thermal shock resistance; usually a smaller value of the Verdet constant as it is impossible to introduce a large concentration of paramagnetic ions due to crystallization; lack of orientational dependence of nonlinear and thermally induced effects and, as a result, the impossibility to influence them. Single-crystal MOEs, in contrast, have an ordered crystal structure and, hence, a significantly higher thermal conductivity, greater strength, thermal shock resistance, as well as better magneto-optical and thermo-optical properties. The ordered structure leads to the anisotropy of the physical and optical properties of the material. Therefore, when manufacturing MOEs it is convenient to use single crystals of cubic syngony, the properties of which are isotropic in the absence of mechanical stresses or strong electric fields, or crystals of a lower syngony–but then the MOE must be cut in the direction of the optical axis [15]. The nonlinear and thermally induced

effects depend on the orientation of the crystallographic axes, hence there must be optimal crystal orientations at which these effects are minimal [16–18]. In any case, this requires greater accuracy when adjusting the MOE, even in the case of cubic crystals. However, single-crystal growth technologies do not allow the growing of high-quality MOEs with apertures comparable to glass ones, as the growth process is much longer and resource-consuming. When growing wide-aperture elements, it is more difficult to control uniformity and obtain samples with high optical quality without internal stresses. The technology of growing single crystals is much more difficult to scale up and does not allow easy transition to mass production. In a number of crystalline materials, it remains possible to change the chemical composition by introducing another chemical element into it– the so-called solid solution or disordered crystals [19]. However, this is not achieved with just any chemical element, nor in any concentration, and the thermal conductivity of such materials is usually lower due to a less perfect crystal structure [20].

Laser ceramics is an ensemble of single-crystal grains, each with its own independent orientation of crystallographic axes, separated by thin grain boundaries (less than 1 nm thick). Ceramic materials combine many advantages of both glass and single-crystal MO materials. In terms of their spectral, optical, and thermal properties, a high-quality ceramic is indistinguishable from a single crystal. At the same time, the technology for the production of optical ceramics, in comparison with the technology of growing single crystals, makes it possible to manufacture optical elements with a larger aperture relatively quickly and relatively inexpensively. Ceramic technology has a great potential for scaling and organizing mass production, greatly simplifies the production of composite optical elements, allows accumulating large active ion concentration, and produces elements with higher uniformity or with a given distribution of the active additive concentration [21]. In the production of optical ceramics, as well as in the production of glass, it is possible to control optical parameters by varying the chemical composition over a wide range. The flexibility of ceramic technology makes it possible to fabricate optical elements from materials, from which it is currently technologically impossible or economically impractical to grow single crystals of comparable quality and size due to the high melting temperature or the characteristics of crystallization and phase transitions of the material. Ceramic technology makes it possible to manufacture ceramics both from a wide variety of materials with cubic crystal lattice symmetry, and from materials with lower crystal lattice symmetry [22]. Methods of producing oriented ceramics from materials with symmetry other than cubic [23,24] and methods that allow converting a ceramic material into a single-crystal one [21,25] are currently underway. This opens up prospects for creating large-size optics of high optical quality from almost any optical material. Due to a large number of ceramic grains on the beam path, the properties that depend on the direction of the crystallographic axes are averaged and the ceramics begins to behave as an isotropic material. This simplifies the adjustment of the MOE from ceramic material, but it loses the orientational dependence of nonlinear and thermally induced effects. The grains of different orientations induce small-scale fluctuations of the polarization [26] and phase of the transmitted radiation [27]. In turn, this increases the scattering of the transmitted radiation [28], reduces the isolation ratio, leads to an additional deterioration of the radiation quality [29,30], and serves as a source of initial noise for the small-scale self-focusing of radiation with high peak and high average power [31]. The difference in the mechanical strength of the grain boundaries and of the ceramic grains themselves and the presence of micropores may lead to difficulties in obtaining a high-quality optical surface with small microroughness when polishing. Such microroughness results in losses due to additional scattering, in a decrease in the breakdown threshold of the surface by high-power laser radiation, and is a source of noise for the growth of small-scale self-focusing. These effects weaken with a decrease in the average grain size in ceramics and an improvement of its optical quality.

## 3. Approaches to the Creation of Faraday Isolators for High-Power Laser Radiation

Let us describe the existing methods and approaches for creating FIs with record characteristics. First, it is necessary to have a transparent material, the manufacturing technology of which makes it possible to obtain an MOE of high optical quality with minimum inclusions and inhomogeneities to minimize "cold depolarization". To reduce nonlinear effects, it is convenient to use large-aperture elements; thus, the corresponding materials technology should be used. Glasses and ceramics are best suited for high peak power lasers as they can be manufactured with apertures larger than 100 mm. When working with radiation with high average power, the main limiting factor is thermally induced effects. The cause-and-effect relationship of the occurrence of thermally induced effects is as follows (Figure 2): the heat release in the MOE due to partial absorption leads to an increase in the average temperature and the formation of a temperature gradient; the temperature gradient in turn leads to thermal and other stresses; the stress due to photoelastic-effect leads to thermally induced birefringence (the direction of eigenpolarizations and the phase incursion between them depend on transverse coordinates and on the amount of heat release). Due to thermal effects, the initially isotropic MOE becomes equivalent to a phase plate with an inhomogeneous cross section, the parameters of which depend on the power of the laser radiation.

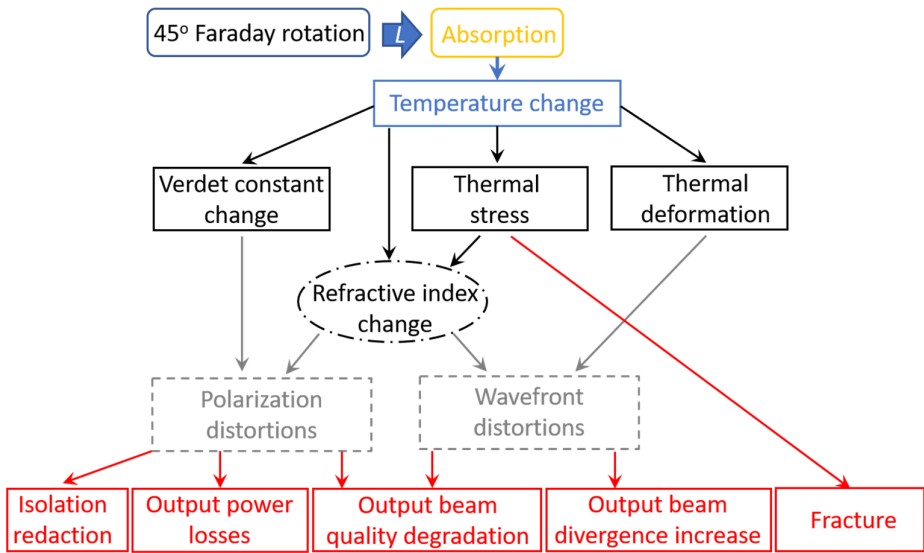

**Figure 2.** A simplified scheme of the physical phenomena involved in the absorption-induced thermal effects in magneto-optical media.

The heat release depends on the absorption coefficient of the material $\alpha_0$ and on the MOE length $L$. By improving the material technology, it is possible to reduce $\alpha_0$. For example, the absorption coefficient of commercial TGG crystals has been reduced by more than four times from 2003 to that date [7]. The MOE length can be reduced by improving the magnetic system and increasing the magnitude of the magnetic field created by it [32,33], or by selecting a MO material with a larger value of Verdet constant $V$. The temperature gradient leads to a transverse dependence of the Faraday rotation angle, which reduces the isolation of the device. To solve this problem, it is possible either to choose a material with $dV/dT = 0$ [34] or to use attenuation methods by magnetic field shaping [35], or schemes with compensation of the temperature dependence of $V$ [36]. Temperature stress is caused by the temperature gradient and thermal expansion of the material. Regions that are warmer in the center press on less heated regions. The temperature gradient can be smoothed out by choosing a material with a high thermal conductivity $\kappa$ or by choosing a proper geometry and method of heat removal [37]. Thermal stresses can be

significantly reduced by choosing a material with a low linear expansion coefficient $\alpha_T$: less expansion ⇒ less pressure of the hotter layer on the colder layer ⇒ potentially less refractive index distortion due to the photoelastic effect. Thermally induced birefringence can be reduced by choosing a material with weak photoelastic properties determined by the coefficients of the piezo-optical tensor $\pi_{ij}$. It is also possible to use the features of the arising thermally induced birefringence to reduce the thermally induced depolarization of the laser radiation. For example, in single-crystal MO materials with a negative piezo-optical anisotropy parameter $\xi$, by choosing a proper orientation, the arising thermally induced birefringence eigenpolarizations may be aligned in one direction and qualitatively reduce the thermally induced depolarization of the laser radiation, and therefore significantly increase the isolation ratio of the FI [17]. When the incident linear polarization coincides with eigenpolarization (that is the same throughout the cross section, with a correct choice of orientation), no polarization distortions will arise in the absence of magnetic field [38]. When all this is insufficient, more complicated FI optical schemes with compensation of thermally induced depolarization can be used: with a reciprocal rotator [39,40], with an absorbing element [41], and with a counterrotation scheme [34,42]. Compensation efficiency is essentially determined by the material parameters. In materials with negative $\xi$, compensation is possible without reciprocal quartz rotator between the MOEs. Of great importance is the mutual orientation of the crystallographic axes in the MOEs [42]. As a consequence, in ceramic materials with $\xi < 0$, due to the random direction of crystallographic axes orientation in the grains, thermally induced depolarization can be partially compensated in the grain-to-grain transition. The efficiency of such compensation increases as the value of $\xi$ approaches $-2/3$. By changing the composition of the material, values of its thermo-optical parameters responsible for thermally induced and magneto-optical properties may be changed ($\xi$, $\pi_{ij}$, $\alpha_T$, $V$) [43]. Thermally induced effects can be significantly reduced by cryogenic cooling, which improves most of the thermo-optical properties of MO materials [44,45] (increases the Verdet constant and the thermal conductivity coefficient, reduces the linear expansion coefficient and the coefficients of the piezo-optical tensor). Each of these methods can be used individually or in any combination, depending on the requirements for isolation, technological simplicity, and the price of the final product. For example, the joined use of optimal orientation and a compensation scheme with a reciprocal rotator makes it possible to significantly reduce (compared to using them separately) the polarization distortions resulting from thermally induced birefringence [46]. It can be seen that the use of most methods requires empirical information about a set of material parameters and their dependence on the MOE temperature, radiation wavelength, and power.

## 4. Comparison and Ranking of Magneto-Optical Materials

It is impossible to find a material in which most of the properties satisfy the above requirements. There arises a natural question about the possibility of introducing a quantitative parameter for comparing MO materials. Historically, the figure of merit (FoM) was first introduced in the form $\mu^* = V/\alpha_0$ [47,48]. The larger the Verdet constant and the lower the material absorption, the higher the FoM and the better the MO are. However, the FoM introduced in this way does not allow a correct assessment of a possible degree of the isolation ratio that a Faraday isolator can provide at a high average power. Some attempts were made to fix this by introducing a FoM, which makes it possible to compare in terms of the minimal introduced thermally induced depolarization (the maximal isolation ratio) at high power $\mu = V\kappa/(\alpha_0 Q)$ [14]. Here, $Q$ is the thermo-optical characteristic responsible for thermally induced depolarization. Later, this parameter was generalized to a wider range of materials and took into account the value of the piezo-optical anisotropy parameter $\xi = \pi_{44}/(\pi_{11}-\pi_{12})$ [7]:

$$\mu = \begin{cases} \left| \dfrac{V\kappa}{\alpha_0 Q} \right|, & |\xi| \geq 1 \\[3mm] \left| \dfrac{V\kappa}{\alpha_0 Q\xi} \right|, & |\xi| < 1 \end{cases} \tag{1}$$

The magneto-optical FoM introduced in this way is inversely proportional to the minimum of thermally induced depolarization in the single crystalline MOE with [001] orientation specified by the photoelastic effect ~$(1/\gamma_{[001]\,min})^{-1/2}$, as for the materials with $\xi > 0$, orientation [001] is the best orientation in terms of the thermally induced depolarization minimum and such introduction of the magneto-optical FoM is quite justified. However, for MO materials with $\xi < 0$, the optimal orientation is [C] which, in general, differs significantly from the [001] orientation [17]. Consequently, the magneto-optical figure of merit introduced according to equation 1 is not suitable for comparing crystalline MO materials with $\xi < 0$. When studying real Faraday isolators, the thermo-optical properties of magneto-active materials are compared using the quantity $P_{max}$ that is the power of laser radiation, at which, due to thermal effects, the isolation ratio reaches a characteristic value–usually $I = -10 \cdot \log(\gamma) = 30$ dB [39,49,50]. This power fully determines the performance of the device and significantly depends on the properties of a particular MO material, the magnetic system, the MOE cooling system used, etc. In the work [7], it is proposed to use $P_{cr}$, $P_{cr,V}$ and $P^*_{max}$ calculated analytically. By equating expressions of the first term of the expansion in a small parameter (polarization distortions $\ll 1$) of integral thermally induced depolarization ($\gamma_{[001]\,min} \approx \gamma_p + \gamma_V$) in the case of single crystaline MOE with [001] orientation to $\gamma_p = \gamma_V = 0.001$ and an expressing power from it can obtain:

$$P_{cr} = \begin{cases} \sqrt{\dfrac{0.016}{A_p}} \dfrac{\lambda B_0}{\pi} \left| \dfrac{V\kappa}{\alpha_0 Q} \right|, & |\xi| \geq 1 \\[4mm] \sqrt{\dfrac{0.016}{A_p}} \dfrac{\lambda B_0}{\pi} \left| \dfrac{V\kappa}{\alpha_0 Q\xi} \right|, & |\xi| < 1 \end{cases}$$

$$P_{cr,V} = \sqrt{\dfrac{0.064}{A_V}} \dfrac{\kappa}{\alpha_0} \left( \dfrac{1}{V} \dfrac{dV}{dT} \right)^{-1}, \tag{2}$$

$$P^*_{max} = \dfrac{P_{cr} P_{cr,V}}{\sqrt{P_{cr}^2 + P_{cr,V}^2}}.$$

where $A_p$ and $A_V$ are numerical factors depended on beam cross-section form factor (for Gaussian beam $A_p = 0.0139$; $A_V = 0.067$). Analytical expressions are obtained in approximations of axial symmetry of the problem, isotropy of the problem of heat conduction and elasticity, in the case of plane deformation. The critical power $P_{cr}$ for the nondecoupling caused by the thermally induced birefringence ($\gamma_p$) and $P_{cr,V}$ is responsible for the nondecoupling caused by the $V(T)$ dependence ($\gamma_V$), so it possible to compare this contribution. The higher the $P^*_{max}$ is and the smaller $\xi$ are, the better the MO medium. However, to obtain the equation 2 the first term used is the expansion of integral thermally induced depolarization in the case of single crystal with [001] orientation, and the obtained critical powers describe the materials with $0 < \xi \leq 1$ and materials with $\xi > 1$ well, but not too well with materials exceeding 1 too much ($\xi \gg 1$). This approach was generalized in work [17], where analytical expressions for the minimum of the integral thermally induced depolarization for materials with $\xi < 0$, $0 < \xi \leq 1$ and $\xi > 1$ taking into account the following expansion orders are obtained. By equating these expressions to 0.001, each can obtain $P_{max}$ value (higher $P_{max}$ better the MO medium). Due to cumbersomeness, these expressions are not presented here. However, to calculate $P_{max}$ it is necessary to know $\alpha_0$, $Q$, $\kappa$, $\alpha_T$, $\xi$, $V$, and $1/V \cdot dV/dT$ that are unknown for new materials and should be measured. Further, to measure some of the parameters, a single crystal element with known crystallographic axes

orientation is needed (for example, $\xi$ and $Q$ [51,52]). As from the point of view of thermally induced depolarization, ceramic elements behave as isotropic materials (glasses), $P_{max}$ may be calculated using the analytical expression for $\gamma$ for a single crystal with [001] orientation at $\xi = 1$ and replacing $Q$ by $Q_{eff} = Q(2 + 3\xi)/5$ [28]. The magnitude of $Q_{eff}$ is calculated from known values of $Q$ and $\xi$, or is measured experimentally [53]. The values of the FoM considered above are presented in Table 1 for a number of MO materials with measured thermo-optical properties. The $P^*_{max}$ and $P_{max}$ are equal for glass, ceramics, and single crystal with $0 < \xi \leq 1$ and can differ significantly for single crystals with $\xi < 0$ and $\xi \gg 1$.

**Table 1.** FoM for MO materials with known thermo-optical properties.

| | $\mu^* = V/\alpha_0$, rad/T | $\mu = V\kappa/\alpha_0 Q$, rad·W/(T·m) | $P_{cr}$, kW | $P_{cr,V}$, kW | $P_{max}$, kW [17] | Experiments $P_{las}$, kW; $I$, dB |
|---|---|---|---|---|---|---|
| MOG04 | 210 | $1.7 \cdot 10^8$ | 0.16 | 2.1 | 0.16 | 0.4; 7 [54] <br> * 0.4; 25 [54] |
| TGG sc | 284 | $7.5 \cdot 10^8$ | 0.7 | 10 | 0.7 | 0.8; 27 [50] <br> * 1.5; 33 [50] |
| TSAG sc | 154 | $320 \cdot 10^8$ | 29 | 3.4 | 1.7 | 1.47; 35 [55] <br> – |
| NTF sc | 320 | $9.1 \cdot 10^8$ | 2.3 | 2.9 | 2.9 | 1.05; 31 [56] <br> – |
| KTF sc | 2267 | $21 \cdot 10^8$ | 1.9 | 32 | 3.0 | 0.32; 42 [57] <br> – |
| TAG cer | – | $4 \cdot 10^8$ | 0.6 | – | 0.6 | 0.4; 33 [58] <br> – |
| $Tb_2O_3$ cer | 60 | $4.5 \cdot 10^8$ | 0.4 | 0.7 | 0.35 | 0.09; 30 [59] <br> – |
| ZnSe @1μm cer | – | $3.2 \cdot 10^8$ | >2.5 | $\infty$ | >2.5 | 1.27; 30 [60] <br> – |
| Si @1.9μm sc | 138 | $3.7 \cdot 10^8$ | 0.53 | $\infty$ | 14 | 0.01; 38 [34] <br> * 0.01; 42 [34] |

Sc-single crystal; cer-ceramics. Estimates made for $B_0 = 2.5$ T; the values of the material constants are taken from [7,17]. The last column contains experimental result of isolation degree at maximum tested laser power in FI traditional scheme and * scheme with compensation.

## 5. Overview of Some Trend Magneto-Optical Ceramic Materials

Rare-earth sesquioxides ($REE_2O_3$) are among the promising MO materials for FIs operating in laser radiation with high average and high peak power. These materials have a high melting point >2400°, most of them have cubic symmetry under normal conditions and possess good mechanical properties, a relatively high thermal conductivity coefficient, and good thermo-optical properties. Many rare-earth ions possess well-pronounced paramagnetic properties; when introduced into different matrices, they lead to large values of the Verdet constant that depends on the type of the ion, its position in the matrix, and the extent of its oxidation; the Verdet constant magnitude increases proportionally to the concentration of the REE ion in unit volume [61,62]. The increase in magnetic ion concentration in glasses is limited by its solubility in the matrix and does not exceed 30 wt.% in the majority of the matrices. In sesquioxide materials compared to garnets, about a three-fold higher concentration of a paramagnetic ion per unit volume could be achieved and, hence, a several-fold increase in the Verdet constant [63]. Recently, magneto-optical properties of a number of sesquioxide metals of lantanoid group were synthesized, including $Tb_2O_3$ [64–66], $Dy_2O_3$ [67,68], $Ho_2O_3$ [69–71], $Er_2O_3$ [72,73], $Yb_2O_3$ [74,75], and $Tm_2O_3$ [76,77]. When analysing the transmission spectra (Figure 1) and magneto-optical properties, we emphasize the following ones:

(1) $Tb_2O_3$ is transparent in the 380–1750 nm range and demonstrates *V* more than three times higher than that of the TGG single crystal throughout the transparency range (3–4 times higher value dependent on wavelength). This material is the most promising candidate for replacing the widely used TGG in Faraday devices operating in radiation of lasers with active elements doped by $Ti^{3+}$, $Nd^{3+}$, and $Yb^{3+}$. The thermo-optical properties of this material were studied at room [78] and cryogenic [79] temperatures. Further, an FI on its basis was developed, and the behavior of its isolation ratio under the action of high-power laser radiation was investigated [59,80]. The main drawback of this material is the presence of a phase transition below the sintering temperature of ceramics and the change in the degree of $Tb^{3+}$ ion oxidation. The first of them may lead to sample destruction at sintering and complicates requirements for the sintering process and its speeds. The second one results in a decrease in the Verdet constant value and in increased material absorption. To date this absorption greatly limits the practical application of $Tb_2O_3$ in FIs for high-power laser radiation.

(2) $Dy_2O_3$ is transparent in the 500–730 nm and 1900–2300 nm ranges. In the 500–730 nm range, it has a 2.5–3 times higher *V* value than the TGG single crystal. In the 1900–2300 nm range, it has rather high values for practical applications (26.5 rad/(T·m) @1900 nm; 20.1 rad/(T·m) @2300 nm [81]). Lasers based on $Tm^{3+}$ and $Ho^{3+}$ ions radiate in this range. Thermal conductivity and linear expansion coefficients of the ceramics based on $Dy_2O_3$ have been measured [82], but other thermo-optical properties of this material have been studied insufficiently so far. No FI in this wavelength range has been demonstrated yet. The drawbacks include difficulties in sintering ceramics of high optical quality of this material. The material demonstrated a sufficiently large extinction ratio, but the transmittance of the obtained samples was far from the theoretical one. Another drawback is a large number of absorption lines in the spectral regions of radiation of widely used commercial lasers based on $Ti^{3+}$, $Nd^{3+}$, $Yb^{3+}$, and $Er^{3+}$ ions, which restricts the applicability of this material.

(3) $Ho_2O_3$ is transparent in the 930–1000 nm and 1300–1700 nm ranges. In transparency ranges it demonstrates a 1.2–1.3 times higher *V* value than the TGG single crystal. The 930–1000 nm range contains the generation lines of lasers based on the $Ti^{3+}$ ion as well as of some diode lasers for $Yb^{3+}$ ion pumping. The 1300–1700 nm range contains the wavelengths of lasers based on $Er^{3+}$ that are used in fiber-optics communication. Ceramics sinter quite well, and samples with transmittance close to the theoretical one have been produced [71]. To date, the authors are not aware of the implementation and study of FI on this ceramic material; its thermo-optical characteristics - thermal conductivity and linear expansion coefficients have been studied [82]. The drawbacks include the presence of absorption lines near the spectral radiation regions of lasers on $Nd^{3+}$ and $Yb^{3+}$ (1030–1070 nm), both in the shortwave and longwave regions. This may lead to a rather large value of $\alpha_0$ at the wavelengths of these lasers and may fully zero the advantages of high *V*. Another drawback is not a very significant superiority over the *V* value of TGG. Some other materials transparent in this wavelength range: TAG [83,84], TSAG [85], $EuF_2$ [86], TZC [87], and others provide a comparable superiority. $Tb_2O_3$ is also transparent in a larger part of this region. This material may only be preferred if it demonstrates good thermo-optical properties at these wavelengths (e.g., small $\alpha_0 \cdot Q_{eff}$) or good magneto-optical properties in the region >2300 nm.

The $Er_2O_3$ and $Yb_2O_3$ sesquioxides have, respectively, a 1.79 and a 1.99 times smaller value of *V* than TGG at the wavelength of 1064 nm, and 4.3 and 5 times smaller *V* than $Dy_2O_3$ at the wavelength of 2100 nm. A specific feature of the $Yb_2O_3$ ceramics is the absence of absorption bands in the 1100–6000 nm region, which indicates that it is possible in principle to obtain a material with very small $\alpha_0$. The $Tm_2O_3$ ceramics has a 2.94 times smaller *V* value than TGG at the wavelength of 1064 nm and a larger number of absorption bands in the visible and infrared ranges. All this makes the $Tm_2O_3$ ceramics an unpromising material. The thermo-optical properties of these materials have not been studied in detail.

Besides the $REE_2O_3$ ceramics, a few other rapidly developing trends in MO ceramics should be mentioned.

(1) MO materials with ferromagnetic properties $REE_3Fe_5O_{12}$ and $REE_3Fe_5O_{12}$ additionally doped with other ions have long been known [88]. One of the most widespread is $Y_3Fe_5O_{12}$ (YIG) that has been historically used as a MO material at the wavelengths >1200 nm, with high transmittance in this range and with a large value of rotation angle per unit length. Fabricating these materials is a rather difficult task. However, relatively recently, ceramic YIG and REE:YIG samples of a large size and relatively high quality [89–91], and $Tb_3Fe_5O_{12}$ samples [92] were produced; there are no principal restrictions on obtaining ceramics of other $REE_3Fe_5O_{12}$. A high value of the specific Faraday rotation angle makes it possible to fabricate a disk MOE for FIs, and a small value of magnetic field saturation allows the use of a compact magnetic systems. The main shortcoming of such materials is the difficulty to obtain an MOE of high optical quality with a small absorption coefficient. In addition, when operating in a magnetic saturation state, the angle of Faraday rotation is determined only by the length of the ferromagnetic MOE that is different for each wavelength and temperature. This complicates the Faraday isolator setup and does not allow tuning to the required wavelength and temperature conditions by simply shifting the MOE into a region of a stronger/weaker magnetic field. Due to a large value of $\mathrm{d}V/\mathrm{d}T$, the contribution to the final isolation ratio of the temperature dependence of the Verdet constant increases substantially. Further, there is also the contribution associated with the domain structure of the material that additionally decreases the isolation ratio of the FI. The thermo-optical properties of such materials have been poorly studied and the authors are not aware of works concerning the behavior of the FI isolation ratio in high-power laser radiation.

(2) MO semiconductor materials with diamagnetic properties. Such semiconductor materials have found wide applications, and the magneto-optical properties of some of them have been studied. A few of them were used for producing FIs for high-power laser radiation. Of the ceramic materials worthy of mention is the ZnSe ceramics used in an FI and studied for the 1940 nm radiation [93]. This material possesses high transparency in a wide wavelength range and a small absorption coefficient. Its drawback is a relatively small value of $V$ that is 3.5 times smaller than of $Dy_2O_3$ in the 1900–2100 nm range, which demands a rather large or complicated magnetic system. Another shortcoming is that it is difficult to obtain samples with a high optical quality. Stresses and inhomogeneities arising in the production of ZnSe ceramics lead to a rather high level of "cold" depolarization and, consequently, a low extinction ratio <30 dB. Another material widely used in microelectronics is silicon (Si). This material is transparent in the 1.1–6.5 $\mu$m range, has a high thermal conductivity coefficient (~150 W/(m·K)), a 1.7 times smaller Verdet constant than in $Dy_2O_3$ in the 1900–2100 nm range, and a highly resistive silicon can have a very small absorption coefficient (4.3 ppm/cm). This material was used for manufacturing an FI and was studied in [34]. The investigation of the Si thermo-optical properties [94] showed that the parameter $\xi = -0.63$, which is quite close to $-2/3$. Therefore, it can be conjectured that, by virtue of the smallness of $Q_{eff} = Q(2 + 3\xi)/5$, a polycrystalline Si will introduce smaller thermally induced polarization distortions than a single crystal in the [001] crystallographic orientation.

## 6. Conclusions

To conclude, ceramic technology enables the fabrication of large-aperture MOEs of high optical quality from materials, from which it is difficult or economically unprofitable to grow single crystals because of the high melting temperature, the presence of phase transitions, or incongruent melting. This technology makes it possible to significantly change the composition of the sintered ceramics and, hence, to control the magneto-optical and thermo-optical properties of the obtained materials in a wider range. It is a rather powerful instrument for conducting research aimed at finding promising MO materials. The MO materials considered in this paper are only a small part of their diversity. The successive search

for materials with a higher Verdet constant value will allow reducing MOE length while retaining the magnetic system. This, in turn, will make it possible to use the disk MOE geometry and to reduce thermally induced effects both due to a smaller length and a more efficient heat sink from the element end [37,95]. With retained MOE length and with the use of a material with a higher *V* value, the magnetic field magnitude can be reduced several-fold, thus significantly reducing the mass and size of the magnetic system due to the logarithmic dependence of the magnetic field value on the ratio of the external and internal radii of the magnetic system [32]. By replacing a MO material by one with a three times higher Verdet constant, retaining the length and diameter of MOE, will allow the reduction in the mass of the magnetic system by about a factor of 26. The study of the properties of new MO materials allows you to correctly compare them with each other. The use of material features will make it possible to choose a material fit for each practical application and to create devices with unique characteristics for any wavelength range.

**Author Contributions:** Conceptualization, methodology, formal analysis, data curation, writing—original draft preparation, visualization, I.S.; supervision, project administration, J.L. All authors have read and agreed to the published version of the manuscript.

**Funding:** This work was supported by the Center of Excellence «Center of Photonics» funded by the Ministry of Science and Higher Education of the Russian Federation, contract No. 075-15-2022-316.

**Institutional Review Board Statement:** Not applicable.

**Informed Consent Statement:** Not applicable.

**Data Availability Statement:** Not applicable.

**Conflicts of Interest:** The authors declare no conflict of interest.

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
