# Peer review of "Selection of Magneto-Optical Material for a Faraday Isolator Operating in High-Power Laser Radiation"

_magnetochemistry, doi:10.3390/magnetochemistry8120168_

Round 1

Reviewer 1 Report

Faraday isolators (laser isolators) are one of the key components of high power lasers. Working at extreme powers any back reflection induced by an optical element surface can damage the early parts of the laser amplification chain. So this work is of interest to a broad audience interested in such high power lasers where it is necessary to use large apertures.

The manuscript is a very clear and clean review of the different options for these Faraday isolators according to the different laser wavelengths you are working with. To my knowledge there is no other similar review in the literature dealing with novel laser ceramics, and it is going to be useful for many scientists in the field, including myself. 

The manuscript is clear and follows a logical trajectory easy to read and understand. 

I do recommend publication

Author Response

Thank you very much for appreciating this manuscript. 

Reviewer 2 Report

The article is well-written, and it's short and easy to understand; only one thing is if the author could add more charts, graphs and tables instead of too much theory, it will be much easy for the readers to understand.

Author Response

Thank you very much for your valuable comments.

Figure 2 and Table 1 were added. 

Reviewer 3 Report

Authors carried out selection of magneto-optical material for a Faraday isolator operating in high-power laser radiation. However, there are certain shortcomings of the article. 

1. The problem description is open-ended. A dedicated stance should be presented. 

2. A comprehensive discussion on the methodology and results is required. The application perspective and limitations could also be highlighted in the abstract to facilitate readers. 

3. A comparative analysis is welcomed. These days the systems are so advanced. Therefore, a detailed analysis on existing literature must be included with the novelty of current study. 

4. At the moment, the paper looks more like a report. The descriptive part should be categorized to literature review/introduction section where you can mention shortcomings of old selection approaches. 

5. A dedicated problem statement, methodology of approach, why the approach is selected, are there quantitative decision support system backing is available?

6. What is reliability of selected approach, how the qualities are being compared? The mechanistic details on the model is missing.

7. The quantitative aspects are missing. Authors should include cases or experiments where specific properties were quantitatively compared. 

Good luck

Author Response

Thank you very much for your valuable comments.

  1. Yes, the discussion has an open character, since the choice of the MO material depends significantly on the parameters of laser radiation, the requirements for permissible distortions and losses, and is limited by the technological capabilities of manufacturing each of the materials today. The calculated value Pmax shows to what power the FI can be used when the requirement for the degree of isolation is met at the other equal conditions. But for new materials, most of the material parameters are unknown, so it will not be possible to compare them correctly. A special position is disclosed in paragraph 4 (paragraph 4 was amended).
  2. This article was written as a review for discussion. It combines already known results and personal point of view of the author.
  3. Thank you for your valuable comment. Links to other reviews of MO materials are added to the text, as well as a table with calculated value Pmax.
  4. I want to disagree. The article is an overview of approaches to the construction of a Faraday isolator with a high degree of isolation in high-power laser radiation in terms of material selection and an emphasis on new trends in the use of magneto-optical ceramics. Describing the pros and cons of each. Also, this work is not a step-by-step guide to the selection of MO material, but it allows to pay attention to the important properties of materials. By choosing materials with better value of these properties or changing these properties by creating new materials it is possible to reduce thermally induced polarization distortions and increase the isolation degree of FI.
  5. “A dedicated problem statement” choice of MO material for FI for high power laser radiation;
    “methodology of approach” measurements of the necessary thermo-optical properties of the material, calculation of thermally induced distortions in the MOE and assessment of the degree of isolation of the device
    “why the approach is selected” it is physically substantiated and agrees well with experimental results
    “are there quantitative decision support system backing is available” yes, Pmax value, calculated at equal conditions for each MO material
  1. Model details added in paragraph 4. The reliability of the approach is due to the implementation of real FIs with a high degree of isolation at high average power and the operation of these devices in real LIGO, VIRGO installations and laser installations of our own use.
  2. The table 1 was added.

Round 2

Reviewer 3 Report

The paper is improved.